# An epifluorescence microscope design for naturalistic behavior and cellular activity in freely moving *Caenorhabditis elegans*

Sebastian N. Wittekindt[1], Hannah Owens[1], Aurélie Guisnet [2],
Lennard Wittekindt[3] & Michael Hendricks [2] ✉

Understanding the neural basis of behavior requires imaging cellular activity in freely moving animals, which typically demands expensive, restrictive microscopy setups. To overcome these barriers, we developed Wormspy, a cost-effective, open-source epifluorescence microscopy system for high-magnification imaging and tracking of *Caenorhabditis elegans*. Wormspy enables the simultaneous recording of neuronal activity and behavioral dynamics without needing the animal to be restrained. We demonstrate its utility in imaging body wall muscles, sensory neurons, and subcellular calcium events within interneuron axons. Our platform reproduces known mutant phenotypes and uncovers, to the best of our knowledge, previously inaccessible sensorimotor correlations. We show that Wormspy provides a robust, modular framework that lowers technical barriers to high-resolution neural imaging, enabling flexible experimental designs for dissecting behavior in freely moving organisms.

The combination of smaller and cheaper imaging technology and tractable model organisms like *Caenorhabditis elegans* promises to be an important driver in our understanding of the neural correlates of behaviour. *C. elegans* is an excellent model organism for imaging-based experiments due to its small size, transparency, and genetic tools[1]. These properties have led *C. elegans* to be widely used in studies of development, cellular functioning, and neural signaling[2], as well as being crucial for phenotyping models of disease and drug development[3]. Its cell lineage is largely invariant, allowing for individual cells to be tracked optically in both wild-type and mutant animals using differential interference contrast microscopy or fluorescent proteins[4]. Specifically, epifluorescence microscopy is a powerful tool for visualizing and studying cell activity, development, and the functioning of the nervous system.

Over the past decade, biological and neuroscience research has seen a marked shift toward custom, open hardware instrumentation, driven by advances in digital fabrication, commodity electronics, and open-source control software that make it feasible for individual labs to design and deploy bespoke systems[5–8]. This movement is not just a cost-saving measure, but a means of enabling task-specific experimental design and improving reproducibility through shared, modifiable platforms. Together, these developments have supported a growing ecosystem of open microscopes, electrophysiology rigs, and behavioral platforms that are increasingly central to modern experimental workflows. In behavioural neuroscience specifically, recent developments in hardware, such as smaller, more sensitive digital cameras, and software, such as segmentation methods based on computer vision models are increasingly making possible the semi-autonomous imaging of single cells, cell assemblies, and even whole nervous systems in freely moving animals.

Imaging in freely moving animals as opposed to restrained animals comes with a spate of engineering challenges, but makes possible new investigations of sensorimotor integration, navigational strategies, and a host of other behavioural research[9,10]. Existing imaging platforms developed to image freely moving *C. elegans* are broadly divided into two categories[11]. Some trackers use a high-resolution camera to image a large area and achieve higher resolution of behavioural and locomotion data by using machine vision software that can

[1]Integrated Program in Neuroscience, McGill University, Montreal, QC, Canada. [2]Department of Biology, McGill University, Montreal, QC, Canada. [3]Independent Researcher, Immenreich, Lindau, Germany. ✉e-mail: michael.hendricks@mcgill.ca

segment and track single or multiple animals[11–14]. These imaging platforms are used almost exclusively for behaviour. Other imaging platforms use high magnification objectives to achieve the resolution necessary to observe cellular processes such as neuronal calcium flux, but are more limited in the behavioural repertoire they can record[15,16].

Systems based on spinning disk confocal microscopes using dual objectives, one for tracking posture measurements and one for volumetric imaging of fluorescent probes, have been used for whole brain imaging in *C. elegans* crawling between an agar substrate and a coverslip[17,18]. In order to resolve individual cells, indicator expression is typically targeted to the nucleus, or cytoplasmic expression is combined with a nuclear marker. While these systems are powerful for imaging cell bodies, several classes of *C. elegans* neurons exhibit characteristic axonal calcium events, which cannot be measured by this approach, either because the indicator is absent from processes or because axons cannot be individually identified in structures like the nerve ring. It is unclear the extent to which animals recapitulate naturalistic behaviours under these experimental conditions. These systems are also very expensive, typically requiring complex customization of high-end base microscopes, and the resulting datasets require considerable computational expertise to analyse and interpret. Finally, to keep the worm centred in the field of view, these imaging platforms typically use XY motors that move the stage, causing sporadic acceleration forces and vibrations, to which *C. elegans* are known to be highly sensitive[19].

We concluded that there was a need for a compact, flexible design that would enable widefield imaging (or photoactivation) of genetically targetable cells in freely moving animals without expensive hardware or complex analysis pipelines. We therefore designed a user-friendly, inexpensive microscope and software package named Wormspy. Wormspy combines single worm tracking with a motor platform that moves the microscope relative to a fixed behavioural arena. 2-channel imaging through a single objective allows for simultaneous recordings of fluorescent signals and behaviour in a single optical path. While we have focused on calcium imaging, Wormspy can be adapted to any number of use cases with the simple exchange of filters and objectives. The modular design of the microscope allows for a high degree of customization to meet experimental needs, such as the inclusion of optogenetic photostimulation or ratiometric recording modules and can be applied to a wide range of experimental setups.

In this work, we demonstrate the applicability of our design to common use cases in *C. elegans* neurobiology. We use Wormspy to characterize the gait parameters and muscle activity underlying a previously described motor coordination phenotype. We show that Wormspy can resolve sensory activity by recording activity in the ASH polymodal sensory neuron as freely moving worms encounter hyperosmotic glycerol barriers. We demonstrate how Wormspy's ability to simultaneously combine manual and automatic tracking makes it suitable for recordings in difficult environments by imaging the activity of the food-sensing AWC[ON] neuron at a lawn border. Finally, we demonstrate the ability of Wormspy to resolve subcellular axonal calcium transients in the RIA interneuron.

## Results

### Microscope design
For a complete description, parts list, and build guide, see Supporting Information. Wormspy consists of a 2-channel fluorescence optical pathway mounted on a motorized stage (Fig. 1A). The modular design allows LED light sources and filter sets to be changed easily. For example, the use of a dual band dichroic mirror allows either simultaneous use of two fluorophores, a single fluorophore and brightfield image, or a photostimulation channel and a recording channel. This configuration was developed principally for calcium imaging applications in single or small numbers of neurons resolvable in the XY plane.

Our configuration uses 470 and 565 nm excitation and two digital cameras, one receiving the emission spectra of GCaMP (502–538 nm), and one receiving the red transmission band from the dual band dichroic. The red transmission was used as a "bright field" image to silhouette the worm against the background for tracking and behavioural analysis, but the addition of an RFP emission filter (603–678 nm) also allowed us to use this channel for ratiometry in the ASH experiment.

Light from two LEDs coupled to excitation filters (in our example, GCaMP and RFP) is directed via a dichroic mirror to a dual band filter cube and reflected onto the sample through a 100 mm tube lens and 7.5x objective lens, which provides a 1440 × 900 µm field of view, just accommodating an adult *C. elegans*. The emission path from the sample passes through the dual bandpass mirror, where it is split with another dichroic mirror and filtered into green and red components to two machine vision cameras.

We used Teledyne FLIR computer vision USB cameras (BFS-U3-23S6M-C) with a sensor optimized for high quantum efficiency (QE) for calcium imaging, and a sensor with at least a frame rate of 10 Hz and a resolution of 1 MP for the brightfield camera (Supporting Information). The software is also compatible with Basler cameras and can be modified to work with other brands that have OpenCV compatible Python libraries.

The optical path is attached to a Z motor stage via a 90° mounting platform. We used a Zaber VSR20A vertical lift stage, which has sufficient torque to smoothly adjust the position of the microscope and keep the subject animal in focus. The Z motor in turn sits on two Zaber TSB60E horizontal translation stages that move the microscope in the XY plane to keep the subject centred in the field of view. The two cameras and the controllers for the three motors are connected via USB to a PC.

### Software design and user interface
For more details, see the GitHub repository linked in the Supporting Information as well as Supplementary Movie 1. Wormspy comes with an open-source software package that includes a user interface written in Python, HTML, Typescript and open-source libraries such as OpenCV and Scikit-Image (Fig. 1B). Once both cameras are connected, the user can specify which camera corresponds to the desired left and right video feeds. If all necessary drivers are installed and the Zaber Launcher application is open to communicate with the stage motors, the program will then load the live feed of the left and right camera.

Once Wormspy is running, the user can enable *ManualMode* to use a dual-joystick controller to move the XY and Z motors to locate the animal and adjust focus. There are three automated tracking modes: (1) thresholding a dark object on a light background, which is most suitable for bright field channels and tracks the centroid of the worm; (2) thresholding fluorescent markers, which tracks the brightest object against a dark background; or (3) DeepLabCut[20] (DLC) integration, which allows users to provide their own pre-trained model for tracking a specific feature like the nose or tail of the animal. Each tracking method calculates the centroid–or feature of interest in the case of DLC–of the thresholded object and makes the necessary conversions for the XY motors to adjust the field of view. This conversion keeps the region of interest centred as the animal moves. On a 100 mm plate, we can track animals for >30 min uninterrupted. For consistent tracking we recommend a minimum frame rate of 5–10 Hz.

Once the animal has been acquired and in focus, the user can also toggle *FocusLock*, which uses the variance of the Laplacian (2nd spatial derivative) to calculate the sharpness of the image from the right camera, and a proportional–integral–derivative (PID) control algorithm to smoothly adjust the Z motor to maintain the desired focus.

The user can specify a project title and directory to save the video recordings. Both cameras can save videos either as "MJPG" compressed video in an audio-video interleaved (AVI) container, which is

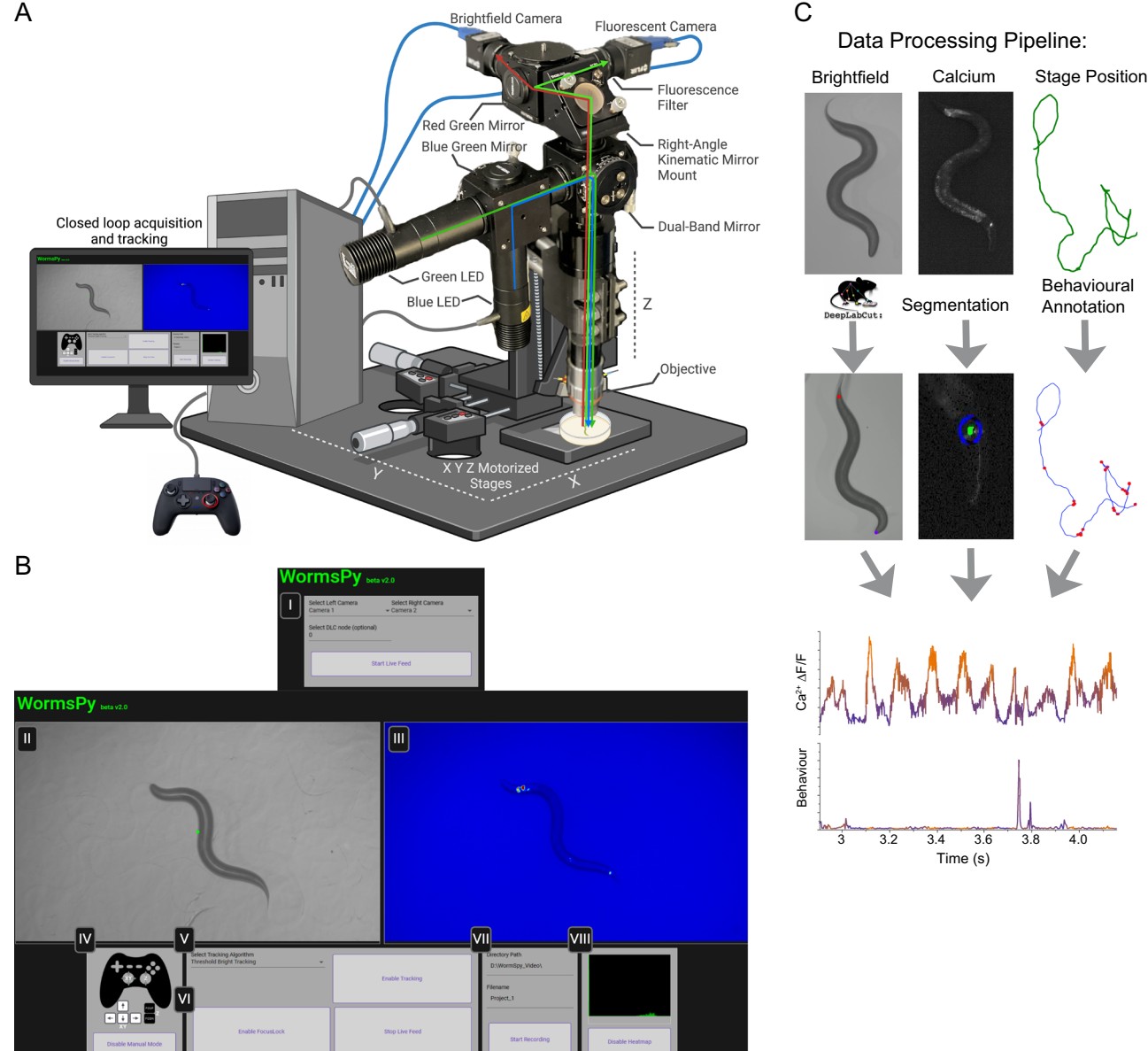

**Fig. 1 | Wormspy hardware, software user interface, and data processing overview. A** Hardware schematic. Green and blue LEDs illuminate the stage through a dual band dichroic filter set, and the sample is captured by two cameras. XYZ motors for automated and manual tracking across a 50.4 × 50.4 mm area, as well as dynamic focus adjustment. Created with BioRender.com. **B** Software UI. (I) Start screen allows users to specify cameras. (II-III) Dual camera feeds. (IV) For manual tracking, a dual-joystick controller is recommended to allow for smooth adjustment of XY for tracking the animal and Z for manual focus correction. (V) Selection of tracking algorithm. (VI) FocusLock adjusts the Z motor to prevent focal drift. (VII) Directory and name to save recording. (VIII) Live histogram for evaluation of fluorescent marker intensity. **C** Example offline data processing pipeline. Brightfield videos can be analyzed to extract pose and behavioural features. Epifluorescence intensity can be analyzed using custom segmentation code. Stage position data can be analyzed to extract behaviourally relevant information, such as position in a gradient and reversals. Frame-synced recordings of brightfield, calcium and stage position data enable easy comparison of behaviour and cell activity.

useful for behaviour, or as a lossless 16-bit tagged image file format (TIFF) stack that preserves image detail and dynamic range. Cameras are frame synced, meaning the two recordings can be directly compared by frame number. A text log of stage position and real coordinates of tracked objects is output for the duration of the recording.

## Calcium imaging in body wall muscle cells during C. elegans locomotion

As our first proof of concept, we used Wormspy to record locomotion and muscle activation dynamics in wild type and uncoordinated mutants. The motor networks that produce *C. elegans* locomotion are modulated by upstream interneurons. We previously reported that disrupting the function of the RIA interneuron via mutation of *gar-3*,

which encodes a muscarinic acetylcholine receptor, is associated with increased body bend amplitude[21] (Fig. 2A).

To test the ability of Wormspy to characterize the effects of *gar-3* mutation on gait and locomotion, we performed calcium imaging in freely moving animals expressing the calcium indicator GCaMP3 in body wall muscle cells (BWMCs) in wild type and *gar-3* mutants. We then analysed the videos with a custom segmentation pipeline (Fig. 2B). In each video, the animal is segmented and a midline constructed from 28 evenly spaced points from nose to tail, and 27 corresponding dorsal and ventral ROIs were constructed from orthogonal line segments (Fig. 2). At each segment, the midline angle relative to the anteriorly adjacent segment was calculated, and the ROIs were used to measure GCaMP intensity in opposing muscle groups in each

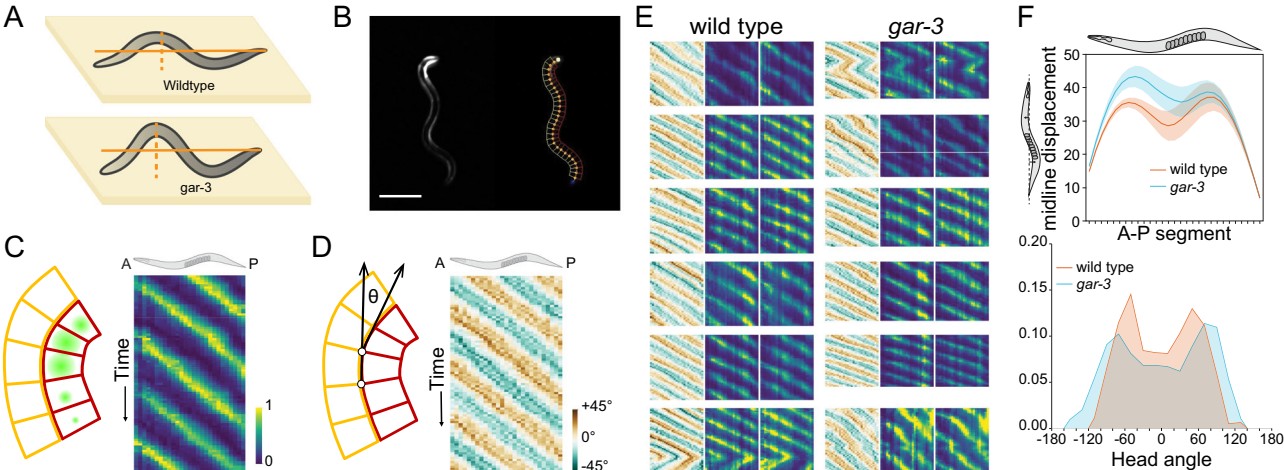

**Fig. 2 | Measurement of locomotion posture and body wall muscle calcium dynamics in freely moving animals. A** *gar-3* mutants display motor coordination defects, including deeper body bends. **B** Example frame of a calcium recording of HBR4 showing GCaMP activation in BWMCs (left) and the segmentation skeleton applied to the worm's midline (right). Scale bar = 250 μm **C** To characterize temporal and spatial distribution of calcium activity in BWMCs, worms were segmented into 28 evenly spaced ROIs by drawing perpendicular lines on either side of the worm's midline (left). Heatmaps show calcium activity from each of these boxes arranged anterior to posterior on the *X* axis and temporally on the *Y* axis (right). **D** For each segment of the skeleton the angle relative to the previous segment was calculated (left) and arranged anterior to posterior as a heatmap (right). **E** Kymographs showing bending angle and right and left calcium activity in wild type and *gar-3* worms. *gar-3* animals show deeper body bends and disrupted coordination. **F** *gar-3* ($n = 12$) worms show increased midline displacement (top) and a wider range of head angles (bottom) compared to wild type ($n = 11$). Shading in top is s.e.m. Data are provided as a Source data file.

segment. We observed the expected calcium dynamics: calcium increased in dorsal BWMCs during dorsal bends and in ventral BWMCs during ventral bends (Fig. 2C, D). Time series analysis for muscle contraction patterns showed that oscillations are slower in *gar-3*, corresponding to a lower gait frequency (Fig. 2E). We confirmed that *gar-3* animals displayed deeper body bends, though only in the anterior half of the body, and showed a broader distribution of head bending angles corresponding to increased gait amplitude, as previously described[21] (Fig. 2F).

## Calcium imaging in the polymodal ASH sensory neuron

To assess Wormspy's ability to resolve single cells and detect sensory-evoked calcium activity, we performed calcium imaging in the polymodal sensory neuron ASH as worms encountered a glycerol barrier. ASH is a ciliated neuron whose dendrite extends to the tip of the nose, where it detects noxious sensory stimuli such as harsh touch, high osmolarity, and aversive odours. Calcium activity in ASH increases rapidly following exposure to these nociceptive cues, enabling reflex-like escape behaviour, including reversals[22,23].

We adapted a behavioural assay by Ghosh et al.[24] to measure the calcium response to a hyperosmotic glycerol barrier in freely-moving animals expressing GCaMP6f and DsRed in ASH (Fig. 3A). To control motion artefacts, we configured Wormspy for ratiometric recording by fitting the second camera with an RFP emission filter. We tracked animals and recorded from when they approached the glycerol barrier until reversal termination. Videos of GCaMP and DsRed signals were analysed using custom segmentation code, which calculates the average of the 25 brightest pixels within the cell soma for both the red and green channels (Fig. 3C). The traces of the green and red channels were then passed through Two-channel Motion Artifact Correction[25] to isolate GCaMP dynamics (Fig. 3B) from shared motion signals. Consistent with previous reports[24], we found that calcium dynamics increased in the ASH soma as *C. elegans* entered the glycerol barrier in the seconds leading up to a reversal, peaking in the initial phase of the reversal (Fig. 3D).

## Calcium imaging in the AWC^ON sensory neuron on food

Tracking and recording calcium activity presents additional challenges when *C. elegans* are on food due to variations in lighting, background,

and focal depth. Wormspy's ability to simultaneously combine manual and automatic tracking modes makes it particularly suited to this task. We tested this capability by recording calcium activity in the AWC^ON neuron as worms exited a patch of food.

The AWC^ON neuron responds to decreases in the concentration of attractive, food-associated chemicals, driving local search behaviour[26]. We therefore expected that we would see an increase in AWC^ON calcium activity as animals exit an *E. coli* lawn. Worms were placed on a small patch of food in the middle of the plate, which was aligned with the midpoint of the Wormspy field of view. This allowed us to convert the relative position of stage movements into the worm's absolute position in the environment (Fig. 4A, C). Our recordings show AWC^ON activity increasing concurrently with the worm's nose leaving the patch of food, consistent with the existing literature[27] (Fig. 4B, D).

## Calcium imaging in RIA axonal compartments

To demonstrate that Wormspy can resolve behaviourally relevant signals in subcellular compartments, we imaged worms expressing GCaMP6f in RIA. As previously described, local, compartmentalized calcium events in the ventral and dorsal segments of the RIA axon (nrV and nrD) are correlated with ventral and dorsal head bends, respectively (Fig. 5A)[21]. The orientation of the axon is such that the nrV and nrD compartments are resolvable in the same focal plane, allowing us to capture the activity in both segments of the axon as worms performed head bends during forward locomotion. Using the brightfield image, we extracted head angles for each frame by plotting points along the animal's midline. We then used this angle to cancel out rotational movement of the neuron in the GCaMP recordings and used Segment Anything Model 2 (SAM2)[28] to define consistent masks for both the nrV and nrD axonal compartments (Fig. 5B). We estimated the local period of head bends for each animal over time. By plotting nrV and nrD activity according to head oscillation period, we show that nrV activity peaks at the most ventral head position, and nrD peaks with dorsal head positions (Fig. 5C, D). Because the motor neuron activity driving local RIA calcium dynamics should be active during bending rather than at the terminus of a bend, we also plotted the derivatives of local calcium domain signals (Fig. 5C, D). Period analysis and cross-correlations showed that the peak rate of calcium increase in each

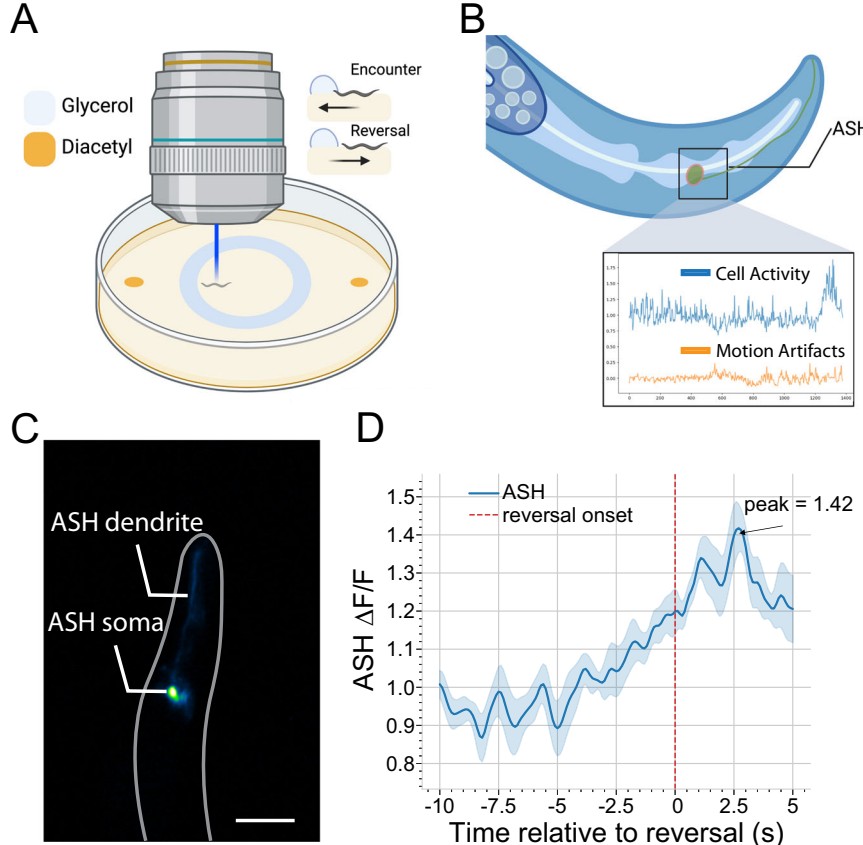

**Fig. 3 | Ratiometric recording of ASH sensory neuron calcium activity in response to an aversive glycerol barrier. A** Illustration of the ASH glycerol barrier assay. Created with BioRender.com. **B** Ratiometric recording of GCaMP6f and RFP allows for the separation of cellular activity from motion artefacts. **C** Example frame of ASH::GCaMP6f animal showing the ASH cell soma and dendrite. Scale bar = 50 μm. **D** Mean and standard error of ASH cell soma fluorescence changes aligned to reversal initiation show sustained activity increases at timescales consistent with the literature in restrained animals ($n = 10$). Source data are provided as a Source data file.

compartment corresponds to peak ventral (positive) or dorsal (negative) head angle velocity. Interestingly, our previous measurements of RIA calcium in restrained worms showed a slightly different periodicity in which peak calcium levels and rates of increase lagged peak head position and velocity. This may reflect the fact that when restrained in microfluidic devices, animals rarely exhibit the regular head oscillations of forward locomotion and spend a large percentage of time in a reversal state, which may involve different timing of motor commands.

## Discussion

The potential exists for imaging technology to be made much more widely available through low-cost, open-source systems. We believe our microscope design is a contribution to this effort. Robust recording of epifluorescence signals and cellular activity has long been difficult in imaging environments that do not incapacitate or restrain the animal. Our design is intended to make fluorescence imaging in freely moving animals as inexpensive and simple as possible, without relying on commercial microscopy systems. While wide field imaging does not allow optical sectioning or resolving cells in the Z-axis, it is more robust with respect to focal movement artifacts, thus common approaches that impede natural movement, like compressing animals between a coverslip and the agar surface are not necessary.

Wormspy was able to resolve the phasic calcium transients of BWMCs and track worms for long intervals to characterize locomotion phenotypes. We imaged the AWC$^{ON}$ neuron as worms exited a patch of food and demonstrated the platform's ability to resolve single neurons in complex imaging environments. Given that both cameras use the same light path and frame-sync recordings, users can align the images

in post. Analysis of both the behavioural data and neural activity collected using Wormspy allows users to make inferences about how neural activity affects behaviour and how the worm's interactions with its environment affect neural activity (Fig. 1C).

Additionally, for more motion sensitive applications, we demonstrated Wormspy's potential for ratiometric recording by imaging the ASH sensory neurons as worms reversed to escape a glycerol barrier. In RIA, we show that it is possible to record calcium activity not only in neuronal cell bodies, but also in dendritic or axonal compartments, provided a sufficiently bright GCaMP line is used. For example, in the RIA::GCaMP6f strain used here, we were able to record with an exposure time of 14 ms, minimizing motion artefacts. Our system allowed us to show how localized calcium dynamics in the RIA axon are correlated with locomotion, identifying temporal differences from data collected in restrained animals. Table 1 provides raw pixel values, dynamic range, and signal to background rations for neuronal GCaMP imaging experiments.

While Wormspy was developed for calcium imaging in head sensory and interneurons in *C. elegans*, its modular design and low cost allow it to be adapted by other researchers and find applications in other fields. Our system is built with off-the-shelf components and is intended to be highly extensible and modifiable. It is particularly amenable to applications like closed-loop optogenetic stimulation. We anticipate that it can easily be applied to other small organisms such as *Drosophila melanogaster* larvae (see Supplementary Movie 2 for a proof of concept recording of Drosophila larvae expressing GCaMP7 in body wall muscle as they perform nociceptive escape behaviour).

Multiple laboratories have developed solutions to image neural activity in freely moving *C. elegans*[16,23,29–34]. While not functionally

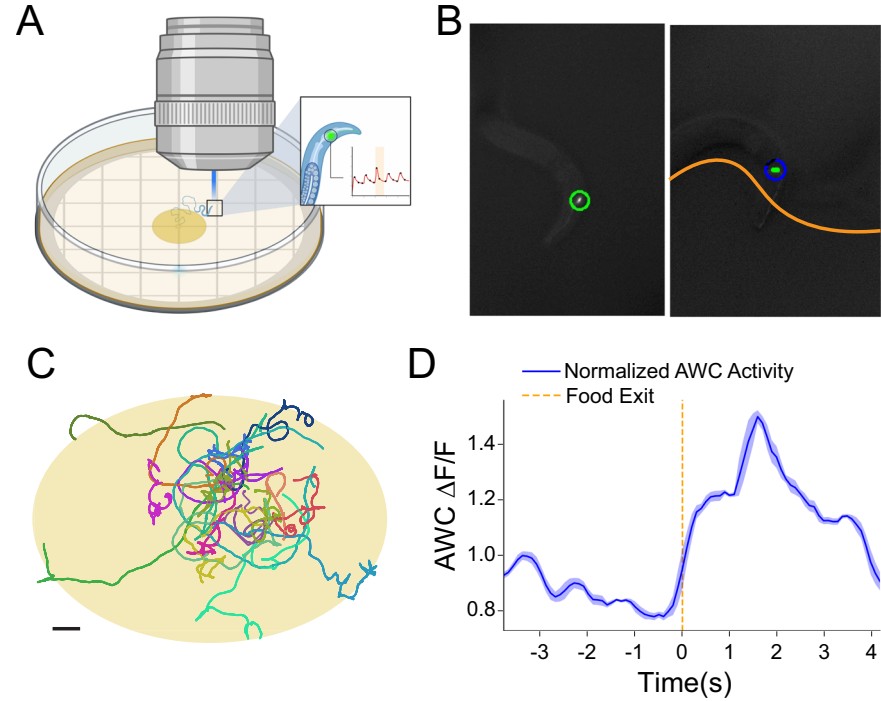

**Fig. 4 | Tracking and calcium imaging of the AWC^ON sensory neuron during food patch navigation. A** Illustration of the AWC food patch assay. Created with BioRender.com. **B** Example segmentation frames of AWC^ON::GCaMP animals navigating on food (left) and with their nose extending out of the food patch (right).

**C** Stage position traces showing worms navigating the food patch assay over a period of 5 min. Scale bar = 1 mm. **D** Mean and standard error of AWC^ON fluorescence response aligned with the time of the worm's nose leaving the food patch (n = 13). Source data are provided as a Source data file.

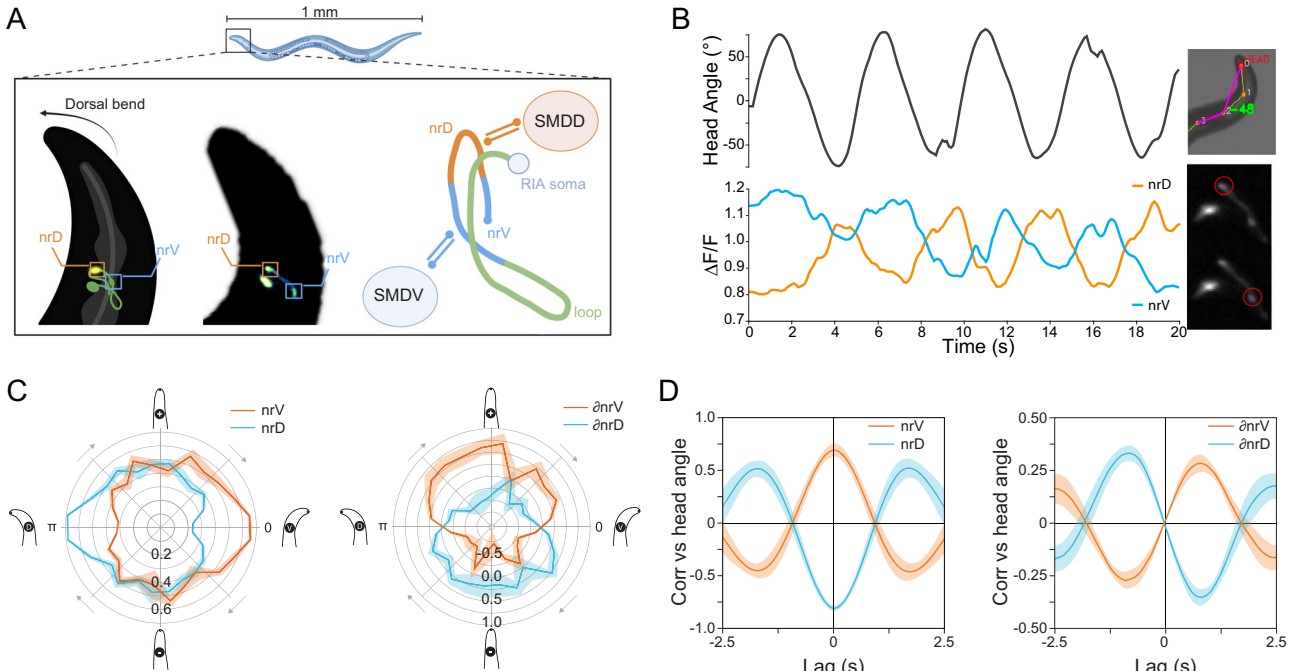

**Fig. 5 | Resolution of localized subcellular calcium events in the RIA interneuron axon during locomotion. A** RIA schematic: the RIA axon exhibits local calcium events in the nerve ring compartments nrV and nrD that encode head movements via reciprocal connections to SMDV and SMDD. Created with BioRender.com. **B** Example trace of head angle alongside nrV and nrD activity over time. Head angle was calculated from points placed along the midline of the worm, and

nrD and nrV were segmented using a custom SAM2 analysis pipeline. **C** Polar plots showing mean and standard error nrV and nrD activity plotted according to oscillation period (left) and the time derivative of nrV and nrD (right). **D** Mean and standard error of cross-correlations of nrV and nrD signal vs head angle (left) and their time derivatives (right) (n = 7). Source data are provided as a Source data file.

**Table 1 | Raw fluorescence signals in imaging experiments.**
ΔF/F is the largest relative change in fluorescence observed, SBR is the signal to background ratio

| Experiment | Median pixel value (neuron) | Mean background | ΔF/F | SBR |
|---|---|---|---|---|
| ASH | 37623 | 1408 | 2.1 | 26.7 |
| AWC | 15540 | 3264 | 0.3 | 4.8 |
| RIA axon | 136 | 15 | 1.6 | 16 |

**Table 2 | Mutant and transgenic C. elegans strains used for behavioral and calcium imaging**

| Strain | Genotype (description) |
|---|---|
| N2 | Bristol wild type |
| VC657 | gar-3 (gk305) |
| HBR4 | goeIs3 [pmec-4::sl1::gcamp3.35::sl2::mkate2-unc-54-3'utr, unc-119 (+)] (BWMC GCaMP3.3) |
| MMH116 | gar-3 (gk305); goeIs3 (HBR4 x VC657) |
| TQ5856 | xuEx1978 Psra-6::Gcamp6 (f);Psra-6::DsRed] (ASH GCaMP6f) |
| AML175 | lite-1 (ce314); wtfIs3 [rab-3p::NLS::GFP + rab-3p::NLS::tagRFP] |
| CX17256 | kyIs722 [str-2p::GCaMP5 (D380Y) + elt-2::mCherry] |
| MMH214 | lite-1 (ce314); kyIs722 (AML175 x CX17256) |
| BAR269 | oleEx103 [(glr-3::GCaMP6f::sl2::RFP) (30 ng/µL pLAU9)]; him-5 (e1490) (RIA GCaMP6f) |

unique, Wormspy adds to this tradition by providing a modular, low cost, open-source hardware and software package to enable combining behaviour and physiological measurements in *C. elegans* and potentially other small organisms. Similar recent work in an accompanying article by Ramahefarivo et al. works toward the same goal of open, modular hardware designs that reduce barriers to designing behavioural neuroscience experiments in *C. elegans*[35]. Wormspy's design, flexibility, and use of standard components make it highly customizable to a wide range of experimental conditions.

## Methods

All experiments complied with relevant ethical regulations. As this study exclusively utilized the invertebrate *Caenorhabditis elegans*, which does not require ethical approval in Canada, no institutional approval was necessary.

*Animals. C. elegans* were maintained under standard conditions and fed OP50 bacteria[36]. To provide age-synchronized young adults for calcium imaging experiments, we used a limited laying technique, transferring the eggs to a new rearing plate where they were raised at 25 °C for 2 days to minimize transgene silencing, before being transferred to 20 °C for 24 h preceding recording. MMH214 was maintained at 20 °C for the duration of development. For measuring BWMC activation, we used HBR4 and MMH116 expressing GCaMP3.3. To measure ASH activity in the glycerol barrier assay, we used TQ5856 animals expressing GCaMP6f. For AWC[ON] recordings, we crossed AML175 with a *lite-1* KO to CX17256 expressing GCaMP5 in AWC[ON] (MMH214). HBR4, AML175, and CX17256 were obtained through the CGC. TQ5856 and BAR269 were generously provided by Shawn Xu and Arantza Barrios, respectively. A list of all strains used in this study is provided in Table 2.

### Statistics and reproducibility

No statistical method was used to predetermine sample size. Sample sizes for the proof-of-concept experiments were chosen based on standard practices and established conventions in *Caenorhabditis elegans* neurobiology and imaging literature, which typically rely on similar cohort sizes to demonstrate robust functional readouts. The sample sizes (biological replicates) for each experimental group were

as follows: $n = 11$ (HBR4) and $n = 12$ (MMH116) for calcium imaging in body wall muscles; $n = 10$ (TQ5856) for calcium imaging in the ASH neurons; $n = 13$ (MMH214) for calcium imaging in the AWCON neuron; and $n = 7$ (BAR269) for calcium imaging in RIA axonal compartments. All videos with robust segmentation were used for the analyses. The experiments were not randomized. The Investigators were not blinded to allocation during experiments and outcome assessment.

### Microscopy system specifications

Complete technical specifications for the Wormspy epifluorescence system, including all optical components, illumination sources, camera details, and acquisition parameters utilized across the subsequent behavioral assays, are detailed in the Supplementary Information and Supplementary Data 1—Light microscopy reporting table.

### Calcium imaging in body wall muscles

We transferred well-fed adult day 1 worms (HBR4 $n = 11$, MMH116 $n = 12$) individually to medium unseeded NGM plates before placing them under the Wormspy objective. The worms were illuminated with blue light and tracked using the user interface. Worms were recorded as they moved across the agar for five minutes. We then analysed the videos with a custom segmentation pipeline to measure ventral and dorsal BWMC activity (Fig. 2B).

### Calcium imaging in the ASH neurons

Young adult TQ5856 ($n = 10$) worms were starved for 30 min. A ring of glycerol (≥99.0%, Sigma-Aldrich) 2 cm in diameter (approximately 12 µL in volume), was pipetted in the centre of medium unseeded NGM plates. To incentivize animals to enter the aversive glycerol barrier, two 1 µL drops of diacetyl (2,3-Butanedione, Sigma-Aldrich, 1:350 dilution) were pipetted outside of the ring, at opposite ends of the plate, about 1 cm from the border (Fig. 3A). The glycerol was allowed to absorb into the agar for ten minutes. Individual animals were then placed inside the centre of the ring and left to crawl freely for five minutes while being tracked and recorded using Wormspy. Blue light was turned on as worms approached the barrier to minimize bleaching. Changes of fluorescence intensity of the calcium indicator GCaMP6f in the ASH neuron were recorded via the GFP-filtered camera. Recordings where the worms encounter the glycerol barrier were segmented for the ASH cell soma via a custom segmentation pipeline (Fig. 3B, C). Frame counts where animals entered the glycerol barrier and initiated reversal were manually ascertained from the brightfield recordings.

### Calcium imaging in the AWC[ON] neuron

Well-fed MMH214 ($n = 13$) worms were individually transferred onto small NGM plates with a small patch of food in the center. Using a grid visible underneath the NGM plates, the plates were aligned such that at the start of a recording, the worm in the patch of food was as close as possible to the midpoint of the Wormspy range of motion (home position). Using a combination of automatic and manual tracking, worms were kept in view and in focus for 5 min as they explored the patch of food. Recordings were manually analyzed to identify frames where the worm's nose exited the food. The brightest 25 pixels of the AWC[ON] cell soma in the GCaMP6f channel were measured for each frame.

### Calcium imaging in RIA axonal compartments

Well-fed BAR269 ($n = 7$) worms were individually transferred onto medium agarose plates without food and recorded using automatic tracking for 1–2 min at a framerate of 10 Hz and 14 ms exposure time to minimize motion artefacts. For each worm, 200 frames of uninterrupted forward runs were selected for analysis. The worm in the brightfield image was skeletonized by placing 12 evenly spaced points along the midline. Head angle was calculated using the angle between the vectors corresponding to points 3→2 and 2→0. The GCaMP6f

channel recordings were cropped into 80 × 80 px frames centered on the neuron soma. Using custom code, we first used head orientation from the brightfield channel to rotationally register each neuron and then prompted the Segment Anything Model 2 (SAM2) to recognize the nrV and nrD compartments and create custom masks that were robust throughout the recording. The brightest 25 pixels in each mask were used for analysis. The traces were smoothed with an equally weighted moving average of window size 0.5 s (5 frames). nrV and nrD calcium were min/max normalized (0–1), and head angles were normalized to the deepest dorsal (−1) or ventral (+1) bend. Derivatives were calculated between adjacent time points on smoothed, normalized values. To estimate the period of oscillation a phase angle was calculated as $\phi = atan2\,(\partial angle, angle)$. Polar plots show mean values within 24 15° bins. For Figs. 5C, D binned by polar coordinates and averaged across worms ($n = 7$).

## Reporting summary

Further information on research design is available in the Nature Portfolio Reporting Summary linked to this article.

## Data availability

The data generated in this study are accessible at https://doi.org/10.5281/zenodo.19477903. Source data are provided with this paper.

## Code availability

https://github.com/Hendricks-Worm-Lab/WormsPy_paper_analysis. The custom software package used for microscope control and tracking (Wormspy) is available on GitHub https://sebzdead.github.io/WormsPy/[37]. The custom segmentation and analysis code is available on GitHub[38].

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

## Acknowledgements

We thank Arantza Barrios and Shawn Xu for strains. Elliott Hendricks assisted in the early stages of software development. A special thank you to Dr. Arjun Krishnaswamy for advice on instrument design. Figures 1a, 3a, 4a and 5a were created in part with assets from BioRender.com. Some strains were provided by the CGC, which is funded by NIH Office of Research Infrastructure Programs (P40 OD010440). This work was funded by grants from the Natural Sciences and Engineering Research Council of Canada (NSERC RGPIN-2020-04928, M.H. and NSERC PGSD 579692, H.O.) and the Canadian Institutes of Health Research (CIHR PJT-155980, M.H.).

## Author contributions

S.N.W. and M.H. designed the instrument. S.N.W., L.W., and A.G. developed code. SNW and HO collected data. S.N.W., H.O., and M.H. analyzed data. S.N.W. and M.H. wrote the manuscript. MH obtained funding.

## Competing interests

The authors declare no competing interests.
