## [Transparent Peer Review file · Nature Communications]

An epifluorescence microscope design for naturalistic behavior and cellular activity in freely moving *Caenorhabditis elegans*

Corresponding Author: Dr Michael Hendricks

Version 0:

Reviewer comments:

Reviewer #1

(Remarks to the Author)

This manuscript by Wittekindt and colleagues discloses WormSpy, a novel epifluorescence microscope setup paired with a customized software solution to simultaneously record behavior and calcium flux in the nematode *Caenorhabditis elegans*. The authors demonstrate the versatility of this platform, using it to assess motor coordination phenotypes, sensory activity of ASH and AWC neurons, and calcium flux in the RIA interneuron, all in free-moving worms. The manuscript is well written, concise, and devoid of any major shortcomings. Given the increasing interest and need to record calcium flux in freely moving worms, this platform will likely see use in several labs and is thus of relevance to the *C. elegans* community. We have only minor comments to offer, which should be straightforward to address.

Minor points:

- 1) The authors write: "For a complete description, parts list, and build guide, see Supporting Information". However, the supporting information only contains links to github and two movies, but no parts list and build guide. Please append.
- 2) Movies S1 and S2 are not referred to in the main text. The link for Movie S2 doesn't work. Based on the description, Movie S2 should show calcium flux in a *Drosophila* larval muscle, an application hinted at but not explicitly explained and/or discussed in the manuscript.
- 3) I wonder if the results shared in Figure 3 could be presented differently. Current Figure 3D does not incorporate positional information of where the worm is at any given time relative to the repellent ring. I'd argue this information should be extractable from the recordings. Ideally, the authors would e.g. compare and disclose ASH activity relative to both time and how close/far the animals are relative to the glycerol ring.
- 4) Figure 4B and C panels are not consistent with the corresponding legends.
- 5) Please increase font size of labels in figure panels 1C, 3B, and 4D.

(Remarks on code availability)

Reviewer #2

(Remarks to the Author)

The manuscript "A novel epifluorescence microscope design and software package to record naturalistic behaviour and cell activity in freely moving *Caenorhabditis elegans*" by Wittekindt and collaborators presents a widefield microscopy system and software to perform brightfield and epifluorescence recordings with freely behaving *C. elegans*.

Since the manuscript mainly concerns advances in methodology and instrumentation, I will likewise focus my review on

these aspects. The authors appear to present their neuroscience results in the second part of the manuscript as broadly in agreement with the existing literature, which I agree with and will not further comment on therefore.

My concern with this manuscript is that it neither presents an advance in the state of the art, nor an ultra-cheap solution that would make *C. elegans* neuroscience widely accessible. It is presumably a useful set-up, defined by a particular budget and for the needs of a specific lab (or type of lab). That is not a problem per se but makes it very difficult to publish. My lab has many such projects, which take many years to complete and become an albatross around our necks – unless we made a genuine methodological breakthrough in some part of the project or a neuroscience discovery.

Here is why I consider this work to not go up against the state of the art: For over a decade now, the field has turned to 3D imaging (confocal or light sheet) to get around the problem of worm neurons overlapping in 2D projection or wide-field imaging. Wittekindt and collaborators are now going back to widefield/2D imaging. Price is clearly a good reason to do so but it does not exactly represent novelty or a work-around for the overlap problem. People were doing calcium imaging in widefield as soon as calcium sensors were invented, sometimes even in freely moving worms. [References are well-known in the field and certainly familiar to the PI.] This is why the authors can only image one or very few neurons at a time.

The authors counter in their introductory paragraph (“Systems based on spinning disk confocal microscopes using dual objectives...”) that neurites cannot be imaged with a confocal, that it is unclear to what extent animals behave naturally in such conditions, etc. Unfortunately, these critiques are either not factually correct (inability of confocal microscope to image neurites) or irrelevant for their case – even if the authors demonstrated these shortcomings of 3D microscopy, they do not offer a sufficiently novel solution compared to what people were doing before. Automated tracking, neural network analysis, and modular optics are all nice features that certainly took serious work to implement, but they exist in different systems at this point already.

Finally, I have doubts regarding some of the details of the set-up. These include the image quality of FLIR cameras, which are in our experience clearly worse and inadequate for calcium imaging in *C. elegans* neurons. Similarly, anything with DeepLabCut and *C. elegans* has been underwhelming.

I completely sympathize with the predicament that I am critiquing, since we find ourselves in the same situation sometimes as well. It is nevertheless my assessment that despite what is clearly nice, useful, and substantial work, this manuscript does not meet the novelty standards usually considered for ‘impact’ and applied for publication in a journal with a broad audience.

(Remarks on code availability)

Reviewer #3

(Remarks to the Author)

Summary

Wittekindt et al. (2025) introduce Wormspy, a compact and affordable epifluorescence microscope system with open-source software for imaging neural and behavioral dynamics in freely moving *C. elegans*. Wormspy integrates dual-channel fluorescence, automated/manual tracking, and a motorized stage to keep animals centered during complex behaviors. The authors demonstrate its versatility by measuring body wall muscle calcium transients during locomotion, sensory-evoked responses in ASH neurons, food-related activity in AWCON, and subcellular calcium events in RIA axons linked to head bending. Wormspy captures known phenotypes (e.g., gait defects in *gar-3* mutants). Overall, the system provides a low-cost, modular platform that expands access to high-resolution neural imaging in unrestrained animals.

At the same time, looking across the literature, it is clear that several groups have already developed their own methodologies to achieve simultaneous calcium imaging and behavioral tracking in freely moving worms. For example, Ravi et al. (2018, *Journal of Visualized Experiments*) established a ratiometric approach using GCaMP5 with mCherry and infrared brightfield tracking to record activity in serotonergic HSN neurons during egg-laying stably. Xue et al. (2025, *Nature Communications*) demonstrated how ASER neuron calcium dynamics encode context-dependent valence by integrating ethanol and NaCl signals, supported by optogenetic manipulations. Nguyen et al. (2015, *PNAS*) developed a powerful imaging system capable of recording whole-brain calcium activity with cellular resolution in unrestrained worms, revealing population-level correlates of locomotor behaviors. Taken together, these studies show that while technical barriers in this field are significant, multiple laboratories have independently overcome them with different solutions. Thus, Wormspy should be viewed as an additional, modular tool that lowers costs and barriers, but not as a unique breakthrough in enabling simultaneous imaging of neural activity and behavior in *C. elegans*. Lastly, several areas need clarification and validation: compatibility with different camera brands, stability of fluorescence-based tracking under dynamic signals, potential frame drops during recording, and a more rigorous description and evaluation of motion correction. In addition, the RIA subcellular imaging data in Figure 5 would benefit from statistical analysis across animals.

Major

1. Could you show that it works for other camera brands as well? Sometimes, it's not as simple as it appears.

“While the software can be modified for any camera type, we used Teledyne FLIR computer vision USB cameras (BFS-U3-23S6M-C).”

2. I have a question regarding option 2, which involves tracking the animal using fluorescence. What happens to the center if the fluorescence intensity changes dynamically over time—for example, with GCaMP imaging? Would the centroid shift along with these fluctuations, and if so, do you recommend brightfield thresholding or DeepLabCut for more stable tracking? Also, could these shifts affect the recordings themselves in any way?

“Once Wormspy is running, the user can enable ManualMode to use a dual-joystick controller to move the XY and Z motors to locate the animal and adjust focus. There are three automated tracking modes: 1) thresholding a dark object on a light background, which is most suitable for bright field channels; 2) thresholding fluorescent markers, which tracks the brightest object against a dark background; or 3) DeepLabCut17 (DLC) integration, which allows users to provide their own pre-trained model for tracking a specific feature. Each tracking method calculates the centroid—or feature of interest in the case of DLC—of the thresholded object and makes the necessary conversions for the XY motors to adjust the field of view. This conversion keeps the region of interest centred as the animal moves. On a 100 mm plate, we can track animals for >30 minutes uninterrupted. For consistent tracking we recommend a minimum frame rate of 5–10 135 Hz.”

3. Did the authors test the fraction of frames that are typically dropped during a recording? If so, what is the observed drop rate in a standard session?

“The user can specify a project title and directory to save the video recordings. Both cameras can save videos either as ‘MJPG’ compressed video in an audio-video interleaved (AVI) container, which is useful for behaviour, or as a lossless 16-bit tagged image file format (TIFF) stack that preserves image detail and dynamic range. Cameras are frame synced, meaning the two recordings can be directly compared by frame number. A text log of stage position and real coordinates of tracked objects is output for the duration of the recording.”

4. Regarding Figure 3B, I have a question about the description of motion correction, which seems somewhat superficial. Could you provide a more detailed explanation of how motion correction is implemented and validated in Wormspy? In particular, it would be helpful to know how you ensure that the correction stabilizes the structural channel while preserving activity-dependent signals, since pitfalls such as over-correction (blurring or stretching), channel misalignment, or residual motion coupled to fluorescence changes are common. Some measurable outputs that could demonstrate effective correction include reductions in framewise displacement and landmark drift in the structural channel, improved ROI stability and footprint consistency over time, increased SNR of activity-dependent signals without distortion of event kinetics, and reduced correlation between motion and activity traces. It would be very useful to know whether you assessed such metrics or could share additional details about your motion correction approach.

5. For Figure 5, there is a mix-up with the calling of the figure in the text. It mistakenly calls Figure 4.

6. For Figure 5, where you show imaging of subcellular events in the RIA axon. While the example traces, polar plots, and cross-correlations are informative, the figure currently reads more as a qualitative demonstration. To convincingly support the claims, it seems important to include some statistical analysis across animals or recordings.

Minor

1. There is a figure legend embedded in the methods (307-312).

(Remarks on code availability)

N/A

Version 1:

Reviewer comments:

Reviewer #1

(Remarks to the Author)

My comments were appropriately addressed and resolved.

(Remarks on code availability)

Reviewer #2

(Remarks to the Author)

I thank the authors for their responses. With regret, I believe my previous criticism of this manuscript stands:

1) Hardware

The authors present a cheaper but standard widefield microscope, replacing parts with less expensive options. Essentially, they avoid a standard microscope body, a standard stage, a standard light source (using LEDs instead), and standard sCMOS cameras, the latter by using cheaper cameras. They cannot get around the cost of standard optics as nobody can

build that on their own. Each of these downgrades comes with tradeoffs, which are not explored (more about this issue below -- especially regarding cameras). Whether anyone can get this system to work is unclear without substantial time investment. If expense is the main argument here, then the time needed to build a custom system versus buying or upgrading an old one should be taken into account as well, e.g., months of PhD student salary and opportunity cost.

Sturdy microscope bodies are strewn around university campuses and cost USD <1k on ebay. The more finicky parts, where old may mean low quality, the motorized x-y stage and cameras, also come with a decrease in quality and durability if they are cheap. Regarding the cameras in particular, we have specific experience with cheaper FLIR high-QE cameras, which came into fashion a few years ago. Their lower quality becomes evident when the GCaMP signal is weak, which is why we ended up having to stick to standard, more expensive sCMOS cameras. I am worried that the authors present standard equipment as not necessary where their solutions may reduce imaging quality in ways that they do not explore at all. One can assemble a cheap microscope or an expensive one, and one tends to get what one pays for. While letting people in the C. elegans neuroscience community know that a cheaper microscope is good enough for some applications, I think the message is too simplistic in the way it has been portrayed.

I would like to address the authors' rebuttals and why they I did not change my mind:

- In response to my previous criticisms, the authors state: "Apologies, an editing error made this section unclear. ... Thus, imaging in neurons with process-specific dynamics (RIA, AIY, RIS, likely others) requires either single-neuron imaging or spectrally separated indicators (rarely used), regardless of imaging modality."

These explanations correct the original manuscript, without addressing my concern -- neither the authors nor anyone else can do dense multi-neuron neurite imaging; thus, this clarification does not address my criticism. They can only image single neurons in widefield -- or multiple well-separated neurons, just like everyone else. There is no technical challenge that the authors' solution overcomes.

- In their next response, the authors state: "We agree that individual components ... This is ... a recognition that experiments that can only be performed by a few specialist laboratories limit the potential of these approaches. Our aim here is open-source instrumentation that is documented and reproducible sufficiently to be used by labs without instrument engineering expertise."

The authors' microscope is even more complicated to assemble than a standard microscope. If someone cannot put together a standard commercial microscope using instructions from the internet, I doubt they can put together a custom microscope. I address software, particularly, the open-source issue below.

- Camera performance

I am glad the authors show data regarding their camera's performance. However, their analysis should have included calcium traces of worms across different ages, including at least L4s - to young adults with different cameras and including individuals that express GCaMP more weakly than others. Furthermore, it should be in the manuscript, not just in the rebuttal letter. Such benchmarks should be made for each of their claims; I should not have to point that out.

2) Software

The authors present the open-source nature of their system as an advantage. However, for example, microManager, a relatively popular open-source microscope control and acquisition software package, has existed for decades, built by the microscopy community over time. I doubt that any single lab, no matter how clever and productive, can produce anything better alone. And if so, the authors should have demonstrated a superior software feature that could not simply have been a plug-in or extension of microManager.

Summary

Again, I have the greatest sympathy for custom designers -- I do recognize the work that has gone into building this custom microscope, but the authors have not shown that their system is superior to existing systems in any respect. The low cost argument is a double-edged sword: Using cheaper parts usually comes with drawbacks (precision, sensitivity, longevity, SNR). None of it is explored by the authors, and I doubt this "result" would suffice for a research article in any case. A standard microscope would cost about the same if one uses the same cheaper light source, cameras, etc. Whether a few months of a PhD salary are worth any potential saving is doubtful. I believe others in the C. elegans neuroscience community would rather just buy a regular microscope (possibly choosing a cheap stage and maybe even the FLIR cameras if these authors can convince them that they are as good), without dealing with the problems that a custom design entails.

(Remarks on code availability)

Reviewer #3

(Remarks to the Author)

In general, the authors have addressed my criticisms. However, I still stand by my point that "Taken together, these studies

show that while technical barriers in this field are significant, multiple laboratories have independently overcome them with different solutions. Thus, Wormspy should be viewed as an additional, modular tool that lowers costs and barriers, but not as a unique breakthrough in enabling simultaneous imaging of neural activity and behavior in *C. elegans*."

The presented data support the conclusion that Wormspy is a functional and thoughtfully engineered system, and it will likely be useful to groups seeking a lower-cost or modular implementation. However, the work primarily positions Wormspy as another tool within an already active and innovative field, rather than as a transformative or field-shifting advance. The core capabilities—simultaneous behavioral tracking and neural imaging—have been demonstrated previously using alternative platforms, and the current contribution appears to lie more in incremental optimization and accessibility than in introducing a fundamentally new methodological concept.

(Remarks on code availability)

Response to reviews

We thank the reviewers for engaging with the manuscript thoroughly and providing their critiques.

Responses to Reviewer #1:

1) The authors write: “For a complete description, parts list, and build guide, see Supporting Information”. However, the supporting information only contains links to github and two movies, but no parts list and build guide. Please append.

Apologies, this was a submission assembly error. The build guide has been appended to the manuscript document.

2) Movies S1 and S2 are not referred to in the main text. The link for Movie S2 doesn't work. Based on the description, Movie S2 should show calcium flux in a Drosophila larval muscle, an application hinted at but not explicitly explained and/or discussed in the manuscript.

We have fixed Movies S1 and S2 in-text references and updated the link. We also added text explaining the significance of recording Drosophila muscles on page 12 of the revised manuscript.

3) I wonder if the results shared in Figure 3 could be presented differently. Current Figure 3D does not incorporate positional information of where the worm is at any given time relative to the repellent ring. I'd argue this information should be extractable from the recordings. Ideally, the authors would e.g. compare and disclose ASH activity relative to both time and how close/far the animals are relative to the glycerol ring.

We do extract exact positional data for the worm, however the precise distribution of the glycerol gradient is not known. Glycerol barriers are manually pipetted onto the agar following a marker guide drawn on the dish, so it can be irregular and diffuses over time. We instead used positional data to determine when the worm was near the glycerol ring and annotated brightfield recordings for reversals that indicate contact with the barrier.

4) Figure 4B and C panels are not consistent with the corresponding legends.

Fixed.

5) Please increase font size of labels in figure panels 1C, 3B, and 4D.

Done.

Responses to Reviewer #2:

My concern with this manuscript is that it neither presents an advance in the state of the art, nor an ultra-cheap solution that would make C. elegans neuroscience widely accessible...

The reviewer details specific critiques supporting this statement, which we summarize and respond to below. On cost, the complete system can be built for \$15K, which is substantially less expensive than any commercial system that performs calcium imaging or behavior alone, and we believe is much less expensive than any previously published custom system that does both. Below, we provide a table of systems used for widefield calcium imaging in freely moving animals, almost all of which rely on base microscopes, optics, cameras, illumination, and software that would cost in the range of \$100-200K. For any system that does volumetric imaging, the cost is substantially higher and requires complex customization.

- 1. The authors counter in their introductory paragraph (“Systems based on spinning disk confocal microscopes using dual objectives...”) that neurites cannot be imaged with a confocal, that it is unclear to what extent animals behave naturally in such conditions, etc. Unfortunately, these critiques are either not factually correct (inability of confocal microscope to image neurites) or irrelevant for their case – even if the authors demonstrated these shortcomings of 3D microscopy, they do not offer a sufficiently novel solution compared to what people were doing before.*

Apologies, an editing error made this section unclear. The contrast here was intended to be with multi-neuronal or whole brain calcium imaging, not with volumetric systems per se. The two approaches used for multilineal imaging are 1) nuclear-localized GCaMPs, where there is no calcium indicator in axons or dendrites, or 2) cytoplasmic GCaMPs with nuclear labels for cell ID, where individual neurites cannot be resolved or identified within fascicles like the nerve ring or nerve cord. Thus, imaging in neurons with process-specific dynamics (RIA, AIY, RIS, likely others) requires either single-neuron imaging or spectrally separated indicators (rarely used), regardless of imaging modality.

- 2. Automated tracking, neural network analysis, and modular optics are all nice features that certainly took serious work to implement, but they exist in different systems at this point already.*

We agree that individual components of our system have precedents in the literature. However, while not universally the case, documentation standards for custom-built instruments have historically been informal and inconsistent. Most systems were described only with schematic diagrams and high-level descriptions, without parts lists, build instructions, or acquisition code. Nearly all prior systems rely on high-end base microscopes, cameras, and often proprietary software.

This is not a critique of this prior work, which has moved the field forward, but a recognition that experiments that can only be performed by a few specialist laboratories limit the potential of these approaches. Our aim here is open-source instrumentation that is documented and reproducible sufficiently to be used by labs without instrument engineering expertise.

Freely moving calcium imaging is not done by the majority of labs who study *C. elegans* behaviour. Papers typically decouple behavioural analysis (in open arenas) and neural imaging (in restrained worms). This is a problem for the field, and the primary barrier is accessible, easily customizable systems. We think there is a broad interest in doing the kinds of experiments Wormspy enables, but technical and cost issues prevent this for many labs.

Below is a best-effort summary of papers that have published widefield calcium imaging data in freely moving animals, focusing on cost/accessibility features:

Reference	Scope	Camera	Software	Build guide	Open source
Ben Arous 2010	n/a	EMCCD (Andor)	LabView	No	No
Faumont 2011	Zeiss Axiovert	sCMOS (Hamamatsu)	Metavue, custom	No	No
Piggott 2011	Zeiss M2Bio	EMCCD (Andor)		No	No
Zheng 2012	Zeiss SV11	EMCCD (Andor)	C, LabView	No	No
Shipley 2014	Nikon Eclipse Ti-U	sCMOS (Hamamatsu)	C	No	Yes
Tsukada 2016	Olympus MVX10	EMCCD (Hamamatsu)	Matlab, Metamorph	No	Partial
Tanimoto 2017	"upright microscope"	EMCCD (Hamamatsu)	Movetr/2D (?)	No	No
Ravi 2018	Zeiss AxioObserver	EMCCD (Hamamatsu)	Zeiss, ImageJ, Bonsai	Yes	Partial

3. *Finally, I have doubts regarding some of the details of the set-up. These include the image quality of FLIR cameras, which are in our experience clearly worse and inadequate for calcium imaging in C. elegans neurons. Similarly, anything with DeepLabCut and C. elegans has been underwhelming.*

Camera performance

At the outset of this project, we assumed that a high-end sCMOS or EMCCD camera might be required for neuronal calcium imaging. However, improvements in GCaMP sensors, less expensive high-power LEDs, and modern machine-vision cameras have changed this tradeoff. FLIR manufactures a large range of cameras with varying sensor types and specifications; the model used here was selected for its high quantum efficiency (we have also used the system with similarly priced/spec'd Basler Ace models). Indeed, the lower-QE FLIR models that we use for wide field behaviour imaging were not adequate for calcium imaging.

Below is a table of raw pixel intensity signal dynamics and signal to background ratios (SBR) for each neuronal imaging experiment:

Experiment	Neuron (median pixel)	Background	$\Delta F/F$	SBR
ASH	37623	1408	2.1	26.7
AWC	15540	3264	0.3	4.8
RIA (axon)	136	15	1.6	16

For imaging cell somata, there was more than sufficient signal under a range of illumination and acquisition conditions (even for the AWC strain, which expresses GCaMP3). Imaging RIA neuronal compartments was more challenging, as each compartment spans only a few pixels in each frame, and exposure time had to be reduced as much as possible to reduce blurring (14ms in these experiments). Nevertheless, SBR remained high and we had no difficulty extracting the previously described characteristic dynamics of the RIA axon compartments.

In summary:

1. Combining volumetric imaging with behaviour requires expensive custom hardware. This is necessary for whole-brain (and most multi-neuronal) imaging experiments. However, as currently performed, only nuclear or somatic calcium is measured, and axon/dendrite imaging cannot be done in multineuronal preps. We do not think that widefield/2D imaging has been supplanted by volumetric imaging, as only a handful of labs do whole-brain imaging, and even fewer do it in moving animals. While this technology has been transformative for the field, arguably the most common behavioral task in *C. elegans* (long-range gradient navigation) has not yet been fully combined with freely moving whole brain imaging because of experimental limitations. Most calcium imaging experiments published in the field use widefield imaging in restrained worms, decoupled from behaviour.

2. Imaging single neurons/processes does not require volumetric imaging and has a wide range of experimental use cases that will be enabled by lower barriers to entry (cost, modular design, user-friendly software, no optical/software expertise required) that many labs face.

DeepLabCut

We agree its application to *C. elegans* is limited—there are few distinct anatomical points on a tube, so within-body markers are unstable. However, the nose and tail of the worm can be tracked robustly. In the majority of cases, contrast-based thresholding in brightfield or fluorescence channels works well. The inclusion of DeepLabCut tracking was intended as an extended option for uses where it might be essential or more applicable: detecting nose and tail in more complex environments, or imaging organisms other than *C. elegans*. The text has been updated to reflect this.

Response to Reviewer #3:

Taken together, these studies show that while technical barriers in this field are significant, multiple laboratories have independently overcome them with different solutions. Thus, Wormspy

should be viewed as an additional, modular tool that lowers costs and barriers, but not as a unique breakthrough in enabling simultaneous imaging of neural activity and behavior in C. elegans.

We agree with the assessment. As discussed above, the intended contribution of this work is enabling these kinds of experiments for labs that do not have the resources or interest in custom instrumentation design by developing an inexpensive system that is fully documented and reproducible. Open-source hardware is an important type of contribution even when it does not rely on new technology.

1. *Could you show that it works for other camera brands as well?*

We have updated the code to incorporate PyPylon to support Basler cameras as well. As noted above, it is important to identify a camera model with as high QE as possible.

2. *I have a question regarding option 2, which involves tracking the animal using fluorescence. What happens to the center if the fluorescence intensity changes dynamically over time—for example, with GCaMP imaging? Would the centroid shift along with these fluctuations, and if so, do you recommend brightfield thresholding or DeepLabCut for more stable tracking? Also, could these shifts affect the recordings themselves in any way?*

Fluorophore tracking (option 2) utilizes a Scikit-Image implementation of Yen's algorithm for binary thresholding. By dynamically calculating the histogram of the image and separating foreground and background, the algorithm responds to changes in fluorophore intensity, provided there is sufficient signal to distinguish it from the background. This makes fluorophore tracking robust for GCaMP imaging, although the dual camera setup makes it possible to revert to backup tracking options such as brightfield or tracking another fluorophore. In any case, changes in cell brightness do not typically change centroid position, or if they do such fluctuations are negligible relative to the size of the worm.

3. *Did the authors test the fraction of frames that are typically dropped during a recording? If so, what is the observed drop rate in a standard session?*

Both cameras use a system of frame queuing that makes it so that frames are captured in real time but only processed when sufficient CPU space is available while preserving frame order. The result is that there are no dropped frames in our experimental datasets. In response to this question, we tested recording both cameras at full resolution and 10Hz for 10 mins while tracking and found that all frames were captured for both cameras. The smoothing algorithm that calculates stage movements means that tracking is insensitive to delayed frame processing over short time scales.

4. *Regarding Figure 3B, I have a question about the description of motion correction, which seems somewhat superficial. Could you provide a more detailed explanation of how motion correction is implemented and validated in Wormspy? In particular, it would be helpful to know how you ensure that the correction stabilizes the structural channel while preserving activity-dependent signals, since pitfalls such as over-correction (blurring or stretching), channel misalignment, or residual motion coupled to fluorescence changes are common. Some measurable outputs that could demonstrate effective correction include reductions in framewise displacement and landmark drift in the structural channel, improved ROI stability and footprint consistency over time, increased SNR of activity-dependent signals without distortion of event kinetics, and reduced correlation between motion and activity traces. It would be very useful to know whether you assessed such metrics or could share additional details about your motion correction approach.*

We used TMAC (Template-Matching and Affine Correction, Creamer et al. 2022, ref 22 in paper), which was specifically developed and validated for *C. elegans* imaging with dual-channel fluorescence. TMAC performs motion correction using the structural channel only, estimating frame-wise affine transformations that correct translation, rotation, and mild deformation relative to a reference template. These transformations are then applied identically to the GCaMP channel (GCaMP), ensuring strict channel alignment and avoiding activity-dependent bias in the correction itself. Because the activity channel is not used to estimate motion, TMAC explicitly avoids over-correction driven by fluorescence transients. TMAC has been previously validated (McWhirter et al., 2022) using quantitative metrics including reduced frame-to-frame displacement, anatomical landmark stability, and preserved ROI shape, and demonstration that neural activity measurements are not distorted. In our data, effective correction is evident from the stable alignment of neuronal ROIs in the structural channel and the absence of motion-locked artifacts in the ratiometric (GCaMP/DsRed) signals. As this method was robustly tested in the original publication and has been used in a number of publications, we did not seek to independently replicate all aspects of its performance. We do note that it qualitatively removed obvious motion effects and resulted in calcium dynamics that matched expectations based on published data. -

5. *For Figure 5, there is a mix-up with the calling of the figure in the text. It mistakenly calls Figure 4.*
Fixed.
6. *For Figure 5, where you show imaging of subcellular events in the RIA axon. While the example traces, polar plots, and cross-correlations are informative, the figure currently reads more as a qualitative demonstration. To convincingly support the claims, it seems important to include some statistical analysis across animals or recordings.*

Apologies, we have corrected the legend to indicate that panels C and D correspond to means/s.d. across 7 animals. The results match the well-described dynamics of the RIA axon.

7. *There is a figure legend embedded in the methods (307-312).*
Removed.

Response to reviewers' comments

Reviewer #1 (Remarks to the Author):

My comments were appropriately addressed and resolved.

Reviewer #2 (Remarks to the Author):

I thank the authors for their responses. With regret, I believe my previous criticism of this manuscript stands:

1) Hardware

The authors present a cheaper but standard widefield microscope, replacing parts with less expensive options. Essentially, they avoid a standard microscope body, a standard stage, a standard light source (using LEDs instead), and standard sCMOS cameras, the latter by using cheaper cameras. They cannot get around the cost of standard optics as nobody can build that on their own. Each of these downgrades comes with tradeoffs, which are not explored (more about this issue below -- especially regarding cameras). Whether anyone can get this system to work is unclear without substantial time investment. If expense is the main argument here, then the time needed to build a custom system versus buying or upgrading an old one should be taken into account as well, e.g., months of PhD student salary and opportunity cost.

*Sturdy microscope bodies are strewn around university campuses and cost USD <1k on ebay. The more finicky parts, where old may mean low quality, the motorized x-y stage and cameras, also come with a decrease in quality and durability if they are cheap. Regarding the cameras in particular, we have specific experience with cheaper FLIR high-QE cameras, which came into fashion a few years ago. Their lower quality becomes evident when the GCaMP signal is weak, which is why we ended up having to stick to standard, more expensive sCMOS cameras. I am worried that the authors present standard equipment as not necessary where their solutions may reduce imaging quality in ways that they do not explore at all. One can assemble a cheap microscope or an expensive one, and one tends to get what one pays for. While letting people in the *C. elegans* neuroscience community know that a cheaper microscope is good enough for some applications, I think the message is too simplistic in the way it has been portrayed.*

We feel that we and this reviewer are inadvertently talking past each other, and perhaps there are differences in disciplinary perspective or approach to experiments. In the field of behavioural neuroscience broadly, making custom instruments from OEM parts and distributors like Thorlabs is entirely standard and is, indeed, necessary for many experiments. Both make high-end

components for custom instruments. Assembly is not considered a barrier, but an opportunity to make the right instrument for the job. The barriers in most labs are familiarity with assembly systems and part names, which is what we try to address by providing a comprehensive build guide, parts list, and software.

The parts used here are not "lower quality" than those used by the big 3 microscope makers, they are just sold without the premium pricing and branding of pre-assembled microscopes. Working in an old "sturdy microscope body" found on campus would not be a benefit here, but a constraint that gets in the way of a well-engineered tracking design and flexibility in other experimental parameters. You can essentially bring Wormspy to any behavioural arena rather than try to cram different behavioural arenas onto a traditional microscope stage.

The reviewer seems particularly concerned about the use of FLIR cameras. FLIR is a brand, not a set of specifications or performance metrics. They are also not "lower quality" than high-end sCMOS cameras, but just different instruments. The differences are known, objective measurements of component performance and specifications, not measures of build quality or reliability. We balanced cost against these specifications and found a point where performance is more than sufficient to our needs and costs are contained. When there are applications that require the use of high-end sCMOS cameras, we use them, but this is not one of them.

The reviewer continues to contrast the effort of assembling a custom instrument with the ease of using a turnkey commercial alternative. It is important to emphasize here that such an alternative does not exist for these experiments.

Given their near ubiquity in behavioural neuroscience, we are unsure of the value engaging in a lengthy justification of why custom instruments are useful and cost effective in a paper like this. There are international conferences on open source microscopy, and entire communities dedicate to open labware. However, We have expanded lines 38-49 to make this context clear.

I would like to address the authors' rebuttals and why they I did not change my mind:

- In response to my previous criticisms, the authors state: "Apologies, an editing error made this section unclear. ... Thus, imaging in neurons with process-specific dynamics (RIA, AIY, RIS, likely others) requires either single-neuron imaging or spectrally separated indicators (rarely used), regardless of imaging modality."

These explanations correct the original manuscript, without addressing my concern -- neither the authors nor anyone else can do dense multi-neuron neurite imaging; thus, this clarification does not address my criticism. They can only image single neurons in widefield -- or multiple well-separated neurons, just like everyone else. There is no technical challenge that the authors' solution overcomes.

We did reply to this concern by pointing out that the vast majority of *C. elegans* papers image single or a few neurons at a time, and combining this approach with behaviour has obvious value. The criticism now seems to be that our system cannot do things that no other system can do, something we never claimed. We provided a bibliography of systems with similar goals and carefully described the advantages ours have over previous efforts. Lines 62-75, 274-277.

- In their next response, the authors state: "We agree that individual components ... This is ... a recognition that experiments that can only be performed by a few specialist laboratories limit the potential of these approaches. Our aim here is open-source instrumentation that is documented and reproducible sufficiently to be used by labs without instrument engineering expertise."

The authors' microscope is even more complicated to assemble than a standard microscope. If someone cannot put together a standard commercial microscope using instructions from the internet, I doubt they can put together a custom microscope. I address software, particularly, the open-source issue below.

The analogy I would use here is "it is more complicated to put together an Ikea desk than to hire a furniture maker to build one and bring it to your house." It is in fact very much more complicated to assemble a standard commercial microscope or a table with turned legs and crafted joints, you are just paying a lot for someone to do it for you. The question is: do you need a system as complex as a the standard commercial microscope, optimized for cell biology and bundled with hundreds of features you don't need and can't opt out of? We have found that 99% of the time, we don't. At the risk of reiterating the point above, in neuroscience the practice of assembling custom imaging instruments from suppliers like ThorLabs is not considered particularly complicated or arduous. Like IKEA, the entire business model is built around it.

Far from wasting a grad student's time, it is, in fact, an excellent skill set for trainees to acquire, and I worry far more about graduate students who use equipment they don't understand than a trainee "wasting" a couple of weeks learning how infinity-corrected fluorescent imaging systems work by assembling one themselves. The idea here is to de-risk that kind of approach for labs that do not do it routinely by providing a pre-validated design and set of components targeted to a common set of experimental contexts.

- Camera performance

I am glad the authors show data regarding their camera's performance. However, their analysis should have included calcium traces of worms across different ages, including at least L4s - to young adults with different cameras and including individuals that express GCaMP more weakly than others. Furthermore, it should be in the manuscript, not just in the

rebuttal letter. Such benchmarks should be made for each of their claims; I should not have to point that out.

This is a new criticism not raised in the initial review, and I disagree that this is necessary in order to show the utility of the system. We used the animal stage that is standard across the vast majority of behavioural studies and multiple strains with varying versions and expression levels of GCaMP. We also successfully imaged individual compartments in the RIA axon that are smaller than larval neurons. I am sure strains exist that express GCaMP so weakly that they would not be usable, but I do not know what objective measure of "expression level" would be suitable for benchmarking against. We feel we have described clearly what the system does and provided examples of it doing so successfully with published strains in standard behavioural contexts. Being asked to describe everything it does not do or every suboptimal use case seems like opening the door to an endless set of low value exercises.

We have now included a summary of pixel values, dynamic range, and signal to background ratios in the paper as Table 1.

2) Software

The authors present the open-source nature of their system as an advantage. However, for example, microManager, a relatively popular open-source microscope control and acquisition software package, has existed for decades, built by the microscopy community over time. I doubt that any single lab, no matter how clever and productive, can produce anything better alone. And if so, the authors should have demonstrated a superior software feature that could not simply have been a plug-in or extension of microManager.

We like μ Manager too, and use it in other contexts. μ Manager and ImageJ have been foundational for microscope control and image analysis, but were developed prior to the widespread adoption of modern scientific Python ecosystems and are primarily optimized for standardized imaging workflows with existing commercial systems rather than tightly integrated, custom instrumentation. As a result, they can be less well suited for applications requiring flexible hardware integration, real-time processing, and closed-loop experimental control, where contemporary Python-based approaches offer greater extensibility and tighter coupling between acquisition, analysis, and device control.

Summary

Again, I have the greatest sympathy for custom designers -- I do recognize the work that has gone into building this custom microscope, but the authors have not shown that their system

is superior to existing systems in any respect. The low cost argument is a double-edged sword: Using cheaper parts usually comes with drawbacks (precision, sensitivity, longevity, SNR). None of it is explored by the authors, and I doubt this "result" would suffice for a research article in any case. A standard microscope would cost about the same if one uses the same cheaper light source, cameras, etc. Whether a few months of a PhD salary are worth any potential saving is doubtful. I believe others in the C. elegans neuroscience community would rather just buy a regular microscope (possibly choosing a cheap stage and maybe even the FLIR cameras if these authors can convince them that they are as good), without dealing with the problems that a custom design entails.

We would argue our system is superior in cost, documentation/reproducibility, and software interface, and is much more user friendly and adoptable by non-specialist labs compared to previously-published systems. Lowering the barrier to performing experiments like this is the primary goal. We have tried to make this clear in the paper. The reviewer's claim about standard microscope costs is off by tens of thousands of dollars, and we would again point out that the components are inexpensive not because the quality is low, but because they aren't sold pre-assembled in a plastic box that says "Zeiss" on it. The savings presented by a system like Wormspy come from not sacrificing quality, but by not paying for the components, feature sets, and associated motorized automation that are bundled with commercial systems (whether you want them or not) and the trust in a brand for service and turnkey operation. These have value if a system does what you need it to and those premium value-added features are important. Plainly, for these experiments, motorized filter wheels and objective turrets, filters and channels that are never used, and \$8000 proprietary control software is not.

Reviewer #3 (Remarks to the Author):

In general, the authors have addressed my criticisms. However, I still stand by my point that "Taken together, these studies show that while technical barriers in this field are significant, multiple laboratories have independently overcome them with different solutions. Thus, Wormspy should be viewed as an additional, modular tool that lowers costs and barriers, but not as a unique breakthrough in enabling simultaneous imaging of neural activity and behavior in C. elegans."

The presented data support the conclusion that Wormspy is a functional and thoughtfully engineered system, and it will likely be useful to groups seeking a lower-cost or modular implementation. However, the work primarily positions Wormspy as another tool within an already active and innovative field, rather than as a transformative or field-shifting advance. The core capabilities—simultaneous behavioral tracking and neural imaging—have been demonstrated previously using alternative platforms, and the current contribution appears to lie more in incremental optimization and accessibility than in introducing a fundamentally new methodological concept.

We agree and have added text to reiterate this in the Discussion (lines 260-).